# Correction of Thin Cirrus Absorption Effects in Landsat 8 Thermal Infrared Sensor Images Using the Operational Land Imager Cirrus Band on the Same Satellite Platform

**DOI:** 10.3390/s24144697

**Published:** 2024-07-19

**Authors:** Bo-Cai Gao, Rong-Rong Li, Yun Yang, Martha Anderson

**Affiliations:** 1Remote Sensing Division, Code 7230, Naval Research Laboratory, Washington, DC 20375, USA; rong-rong.li.civ@us.navy.mil; 2Department of Forestry, Mississippi State University, Mississippi State, MS 39762, USA; yy285@msstate.edu; 3USDA ARS, Hydrology and Remote Sensing Laboratory, Beltsville, MD 20705, USA; martha.anderson@usda.gov

**Keywords:** remote sensing, sensors, atmosphere, cirrus clouds, Landsat 8, surface temperature

## Abstract

Data from the Operational Land Imager (OLI) and the Thermal Infrared Sensor (TIRS) instruments onboard the Landsat 8 and Landsat 9 satellite platforms are subject to contamination by cloud cover, with cirrus contributions being the most difficult to detect and mask. To help address this issue, a cirrus detection channel (Band 9) centered within the 1.375-μm water vapor absorption region was implemented on OLI, with a spatial resolution of 30 m. However, this band has not yet been fully utilized in the Collection 2 Landsat 8/9 Level 2 surface temperature data products that are publicly released by U.S. Geological Survey (USGS). The temperature products are generated with a single-channel algorithm. During the surface temperature retrievals, the effects of absorption of infrared radiation originating from the warmer earth’s surfaces by ice clouds, typically located in the upper portion of the troposphere and re-emitting at much lower temperatures (approximately 220 K), are not taken into consideration. Through an analysis of sample Level 1 TOA and Level 2 surface data products, we have found that thin cirrus cloud features present in the Level 1 1.375-μm band images are directly propagated down to the Level 2 surface data products. The surface temperature errors resulting from thin cirrus contamination can be 10 K or larger. Previously, we reported an empirical and effective technique for removing thin cirrus scattering effects in OLI images, making use of the correlations between the 1.375-μm band image and images of any other OLI bands located in the 0.4–2.5 μm solar spectral region. In this article, we describe a variation of this technique that can be applied to the thermal bands, using the correlations between the Level 1 1.375-μm band image and the 11-μm BT image for the effective removal of thin cirrus absorption effects. Our results from three data sets acquired over spatially uniform water surfaces and over non-uniform land/water boundary areas suggest that if the cirrus-removed TOA 11-μm band BT images are used for the retrieval of the Level 2 surface temperature (ST) data products, the errors resulting from thin cirrus contaminations in the products can be reduced to about 1 K for spatially diffused cirrus scenes.

## 1. Introduction

The Operational Land Imager (OLI) and the Thermal Infrared Sensor (TIRS) instruments are both onboard the Landsat 8 and Landsat 9 satellite platforms [1,2]. A cirrus detection channel (Band 9) [3] centered within the 1.375-μm water vapor absorption region was implemented on the OLI to assist with cloud masking. This band is very sensitive for remote sensing of cirrus clouds from space at a high spatial resolution of 30 m. The TIRS instrument has two IR bands centered near 11 and 12 μm, respectively. Since April 2013, the Landsat 8 Level 2 Surface Products (L2SPs), including surface reflectance (SR) and surface temperature (ST), have been publicly released by U.S. Geological Survey (USGS). The Level 1 top of atmosphere and spatially registered products are referred to as L1TPs. ST is an important geophysical parameter in global energy balance and hydrological research. The ST data products from different sources have been used for monitoring crop and vegetation health [4], wild fires, volcanic eruptions, and urban heat islands, which are resulted from absorption of solar radiation by pavement, buildings, and other surfaces and retaining the heat by manmade materials.

The Landsat 8/9 ST products are generated with a single-channel algorithm developed by the Rochester Institute of Technology (RIT) and NASA Jet Propulsion Laboratory (JPL) [5]. The ST retrievals are made from the L1TP data products, including the 11-μm band brightness temperature (BT), TOA reflectances, and additional ancillary input data such as normalized difference vegetation index (NDVI), surface emissivity [6], and atmospheric profiles from other sources. The possible presence of cold thin cirrus clouds located at the upper part of the troposphere and lower part of the tropopause is not taken into consideration during operational retrievals. Thin cirrus clouds can absorb IR radiation originated from the Earth’s surfaces and re-emit IR radiation at lower temperatures.

The developers of the Landsat ST algorithm estimated that the errors in the temperature products are typically about 1 K from the earlier Landsat 4, 5, and 7 data to the present Landsat 8/9 data. Through global comparisons between Landsat 7 STs and those obtained from data acquired with the NASA MODIS (Moderate Resolution Imaging SpectroRadiometer) Instrument [7,8], it has been found that the differences between the two products are generally less than 1 K [9]. By large-scale comparisons of in situ-measured lake water temperatures against the Landsat 5/7-derived temperatures for carefully selected clear scenes, it is found that the absolute mean difference is 1.34 K for lake pixels 180 m from land. The mean difference for pixels at the lake water boundary areas is 4.89 K [10]. Intercomparisons between STs retrieved from other satellite data acquired under very clear atmospheric conditions and in situ-measured temperatures have also shown consistencies of about 1 K [11,12].

The Landsat series of satellites continue the collection of multi-band imagery of the Earth’s surfaces at a spatial resolution of 30 m. At such a scale, natural and human-induced changes can be detected [1]. The Landsat data together with coarser spatial resolution data, such as those of MODIS at a spatial resolution of 1 km are widely used by researchers for estimating evapotranspiration (ET) [4], the exchange of water vapor between the land and atmosphere. Daily ET maps at a 30 m resolution from Landsat are routinely generated under the OpenET project in support of water management applications, vegetation condition monitoring, drought detection, yield prediction, and irrigation scheduling [13]. Land surface temperature is a key input for ET retrieval using surface energy balance methods. The use of Landsat ST data products for ET estimates can sometimes result in unusually large or small ET values due to thin cirrus contamination. 

We previously developed an empirical and effective technique for removing thin cirrus scattering effects in OLI images [14]. This technique makes use of the correlations between the 1.375-μm band image and images of any other OLI bands located in the 0.4–2.5 μm solar spectral region. In this article, we describe a variation of this technique, i.e., using the correlations between the Level 1 1.375-μm band image and the 11-μm band BT image, for the effective removal of thin cirrus absorption effects in the 11-μm band BT image. In Section 2, we provide more background information about cirrus contamination effects in Landsat 8/9 data products, describe OLI and TIRS spectral properties, atmospheric profiles and cirrus absorption and scattering properties, the empirical technique for removing thin cirrus effects from OLI bands located in the 0.4–2.5 μm solar spectral range, and the extension of this technique for the empirical correction to the TOA brightness temperature (BT) image of the 11-μm band. We present in Section 3 sample corrections to cirrus-affected BT images for three spatially homogeneous cases, and we also highlight the problems of pixel shifts between the 11-μm band image and the 1.375-μm band image that hinder the corrections for spatially non-uniform cirrus cases. We provide brief discussions in Section 4 and concluding remarks in Section 5. We certainly expect that if the cirrus-removed TOA 11-μm band BT images are used for the retrieval of the Level 2 surface temperature (ST) data products, the errors resulting from thin cirrus contaminations in the ST products can be reduced significantly.

## 2. Background and Methods

### 2.1. Background on Cirrus Contamination Effects in Landsat 8/9 Data Products

Figure 1 is an example for illustrating the thin cirrus effects on the Landsat 8 ST data products over land. Figure 1A is a portion of a true color image generated from the red, green, and blue band (RGB) data contained in a Landsat 8 L1TP data set, which was acquired on 1 July 2018, over the eastern coastal areas of Maryland State, USA. Figure 1B is the black/white (B/W) TOA cirrus band reflectance image (1.375-μm band), where black corresponds to zero reflectance and bright white to 0.05 reflectance. In general, the thin cirrus clouds seen clearly in the Figure 1B 1.375-μm band image are not easily identifiable in the Figure 1A RGB image. The thin cirrus-affected areas in the upper left portion of Figure 1A are faint, but clearly impact the ST product (Figure 1C). Cirrus-affected areas are generally cooler than the nearby clear land areas. In order to demonstrate more obviously the thin cirrus effects on Landsat 8 ST data products, we show in Figure 1D the ST image in false color. The ST values over cirrus-affected areas (deep blue color) are about 20 K lower than nearby clear areas (red color). By comparing Figure 1B with Figure 1D, it is seen that thin cirrus clouds can introduce errors of ~20 K into the Landsat 8 ST data products over land. Such cirrus-induced ST errors are much larger than the errors (about 1 K) introduced from all other sources (e.g., atmospheric profiles) during the operational generation of the Landsat ST data products at USGS.

The Landsat 8 ST data products over water bodies can also be contaminated by thin cirrus effects. Figure 2 shows such an example. Figure 2A is a part of an RGB image processed from an L1TP data set (Raw: 001, Path: 085) acquired on 5 March 2014, over the Pacific Ocean west of the coastal area of Chile. This is a thin cirrus scene we previously identified and used during the development of an empirical technique for removing thin cirrus effects in OLI visible, near-infrared (NIR), and shortwave IR (SWIR) spectral regions. Figure 2B is the B/W image of the 1.375-μm cirrus band, where black corresponds to zero reflectance and white to 0.03 reflectance. Thin cirrus clouds are seen in both the Figure 2A,B images. Figure 2C is the Level 2 surface reflectance image, where the thin cirrus clouds are also clearly seen. This indicates that over the dark water surfaces, the thin cirrus scattering effects are propagated down to the Landsat 8 Level 2 SR data products generated with version 1.5.0 of the Land Surface Reflectance Code (LaSRC) [1,2]. Figure 2D is the false-color-coded Level 2 ST image with a color bar placed right to the image. Thin cirrus covered areas have smaller ST values. The largest ST differences between cirrus pixels (deep blue) and nearby clear pixels (red) are approximately 12 K in the Figure 2D image. The images in Figure 1 and Figure 2 have demonstrated that thin cirrus clouds can introduce errors up to 20 K in the Landsat 8 ST data products. In order to increase the utility of the ST data products, such as routine ET estimates over land surfaces and operational monitoring of water surface temperatures, the thin cirrus absorption effects need to be properly corrected for.

### 2.2. The Landsat 8/9 OLI and TIRS Instruments

The Landsat 8/9 OLI and TIRS instruments have a total of 11 bands. OLI has 9 bands located in the 0.4–2.5 μm solar spectral region, while TIRS has 2 thermal IR bands. Their names, positions, and widths are listed in Table 1. The designers of OLI Band 9, which is officially referred as the “cirrus band” by the Landsat Project, adopted the same specifications in band position, width, and saturation radiance as those specified by us for MODIS in late 1992. This OLI band is very sensitive for remote sensing of cirrus clouds from space at a high spatial resolution of 30 m, as shown in Figure 1B and Figure 2B. This band detects thin cirrus clouds from satellite against a nearly black surface background, because the solar radiation in this band transmitted through cirrus clouds in the downward sun-surface ray path is almost completely absorbed by atmospheric water vapor beneath cirrus clouds. As a result, this band receives only the cirrus scattered solar radiation in the sun–cirrus–satellite ray path.

### 2.3. Cirrus Absorption and Scattering Properties

In order to have better understanding of cirrus clouds, we show in Figure 3 two vertical atmospheric temperature profiles, for the Tropical and the US Standard (1976) model atmospheres [15]. For the green-colored 1976 US Standard model curve, the troposphere extends from 0 to 11 km. As altitude increases, the temperature decreases at a rate of about 6.5 K per kilometer. The thick tropopause layer, where the atmospheric temperature remains constant, extends from 11 km to 20 km. The stratosphere, where the atmospheric temperature increases with increasing altitude, extends from 20 km to approximately 45 km. The establishment of this standard model atmosphere and the characteristic shapes of the vertical temperature profile are well described in a scientific document [16]. For the red-colored Tropical model temperature profile, the troposphere extends from ground to about 16 km, which is about 5 km higher than the corresponding US Standard (1976). The increased air temperature below 10 km in the Tropical model causes a stronger atmospheric convection, and results in a higher tropopause altitude (approximately 17 km).

Cirrus clouds are generally formed at the upper part of the troposphere (approximately 10 km) and the lower part of the tropopause, where the air temperatures are very low (approximately 220 K). When the warm and moist air near Earth’s surfaces is pushed upward and cooled down near 10 km, cirrus clouds are generated. Our previous simulation study [17] has demonstrated that a thin layer of partially transparent cirrus clouds with an optical depth of 1.0 in visible spectral regions and with a large effective ice particle size of about 100 μm can lower the TOA BT values for narrow bands centered near the 4.65 μm atmospheric window region by approximately 20 K. Another theoretical study [18] has shown that for a similarly semi-transparent thin cirrus layer with an optical depth of 1.0 in visible spectral regions, the TOA BT values for bands near 11-μm can also decrease by about 20 K. Based on multi-year research about cirrus optical properties [19] and on the development of operational MODIS cloud retrieval algorithms [20,21], it can be inferred that the reflectances of a thin cirrus layer with an optical depth of 1.0 in visible spectral regions can vary between about 0.025 and 0.1, depending on the mean ice particle sizes, the solar and viewing angles. It should be pointed out that the properties of thin cirrus clouds with optical depths less than 2.0 in the visible spectral regions are not retrieved with the MODIS algorithms [20,21], although the thin cirrus clouds have significant effects on temperatures and reflectances of dark surfaces (as shown in Figure 1 and Figure 2).

In Figure 4, we show a diagram further illustrating the mechanisms by which thin cirrus clouds can have large effects on satellite measured brightness temperatures (BTs) near 11 μm. The IR emitted radiances originating from earth surface are absorbed, scattered, and re-emitted at much lower temperatures (approximately 220 K) by cold cirrus clouds at high altitudes (approximately 10 km). The downward solar radiation at 1.375-μm from space is scattered backup by cirrus clouds. The transmitted 1.375-μm radiance is absorbed by water vapor below cirrus clouds. As a result, the 1.375-μm band detects cirrus clouds against a ‘black’ surface background. 

### 2.4. Removal of Cirrus Effects in Landsat 8 OLI Images for Bands in the 0.4–2.5 μm Range

As stated earlier, we previously developed techniques for cirrus detection and cirrus correction from hyperspectral and multi-spectral imaging data [22,23,24]. The MODIS version of the algorithm was made publicly available from a NASA data center more than ten years ago. As far as cirrus reflection and scattering effect is concerned, we can assume that a homogeneous thin cirrus layer is located above a “virtual surface”, which includes the effects of scattering by molecules, aerosols, and low clouds as well as land or ocean surface reflection and sub-surface scattering. To facilitate the descriptions, we omit the wavelength (λ) dependency, and denote the “apparent reflectance” at the satellite for a given band as
ρ* = π L/(μ_0_ E_0_)(1)
where L is the radiance measured by the satellite, μ_0_ the cosine of solar zenith angle, and E_0_ the extra-terrestrial solar flux. In general, ρ* consists of the cirrus reflection component (ρ_C_) and the virtual surface reflection component (ρ_S_), i.e.,
ρ* = ρ_S_ + ρ_C_(2)

Because the 1.375-μm band receives only the scattered solar radiances without underneath atmosphere and surfaces, we utilize an empirical scatter plot to transform the cirrus information in the 1.375-μm band to a given band that has the underneath surface reflectance component. Below, we use a specific Landsat 8 OLI scene to illustrate the cirrus removal technique for bands in the 0.4–2.5 μm solar spectral range.

Figure 5A is a portion of a Landsat 8 OLI RGB image acquired on 21 March 2014, over the Pacific Ocean west of Chile. Spatially homogeneous thin cirrus clouds are seen in the upper left portion of the image. Figure 5B is the 1.375-μm cirrus band image. Figure 5C is the scatter plot for the cirrus band image versus the 0.86-μm band image. Cirrus pixels are clustered around a straight line (in red color). The slope of this line, S_0.86_, can be estimated with a technique described in [23]. The cirrus-removed virtual surface reflection component for the 0.86-μm band image can be expressed as
ρ_S0.86_ = ρ*_0.86_ − ρ*_1.375_/S_0.86_(3)

Figure 5D is the cirrus-removed RGB image. Cirrus features seen in Figure 5A,B are properly removed in Figure 5D, while the surface water waving patterns in Figure 5A are preserved in Figure 5D.

The procedures for removing cirrus scattering effects for a given solar band (B) using the correlations between this band and the 1.375-μm cirrus detection band have been previously described [24]. A total of five steps are involved: (a) converting the TOA measured radiances (L) of Band B into TOA apparent reflectances, ρ*(B), based on Equation (1); (b) making the scatter plot ρ*(1.375-μm) versus ρ*(B) (for illustrative purposes, see Figure 5C); (c) obtaining the slope, S_B_, from the scatter plot as illustrated in Figure 5C; (d) calculating the cirrus reflectance image, ρ_c_(B), for Band B, which is equal to ρ*(1.375-μm)/S_B_; (e) subtracting out ρ_c_(B) from ρ*(B) to obtain the cirrus-removed band B virtual surface reflectance image, ρ_s_(B). As will be seen shortly, a slight variation of this technique is very effective for the removal of thin cirrus absorption effects in Landsat 8/9 OLI 11-μm band BT images.

### 2.5. The Linear Relationship between 1.375-μm Band Cirrus Scattering and 11-μm Band Ice Absorption

We show in Figure 5E the 11-μm band TOA BT image for the same thin cirrus scene of Figure 5A. The areas obviously affected by absorption of ice particles within cirrus are dark. Figure 5F is the false-color-coded BT image with a color bar placed below the image. The blue- and green-colored areas are affected more by ice absorption effects than the red-colored areas. Figure 5G is the scatter plot of 1.375-μm band image versus the 11-μm band BT image. A linear relationship between the two quantities with a negative slope is observed. The slope value, *S*_BT_, can be estimated numerically with the same procedure, as described in [23,24]. Subsequently, the cirrus absorption effect in the 11-μm band BT image can be removed according to
BT_Cir_Corr_(11-μm) = BT(11-μm) + |*S*_BT_| × *ρ**(1.375-μm),(4)
where |*S*_BT_| stands for the absolute value of *S*_BT._

The method described in Equation (4) for removing cirrus absorption effects in the OLI 11-μm band TOA BT images is applicable for scenes with spatially homogenous thin cirrus coverage. It is not practically suited to perform cirrus removal for OLI scenes having rapid variations of thin cirrus in the spatial domain. For examples, the cirrus features bounded inside the small rectangular red boxes in the lower right portion of Figure 1B,C (as well Figure 1D) are relatively shifted by approximately 8 pixels in the horizontal direction and 18 pixels in the vertical direction. Similarly, the cirrus features bounded inside the red-colored rectangular boxes in Figure 2B,C (or Figure 2D) are relatively shifted by about 12 pixels in the horizontal direction and 18 pixels in the vertical direction. The spatial displacements of cirrus features in different OLI images are due to the parallax effect, which is associated with the intrinsic designs of the OLI focal plane assembly [1,25,26].

## 3. Results

In this section, we present three homogeneous thin cirrus case studies on removing cirrus absorption effects in the 11-μm band BT images. The selection of spatially diffuse thin cirrus cases alleviates the problems in removing cirrus absorption effects due to mis-matching of cirrus spatial features in different bands, as discussed in Section 2.5.

### 3.1. A Water Scene West of the Coastal Area of Chile, 21 March 2014

The Landsat 8 OLI RGB image acquired on 21 March 2014, over the Pacific Ocean west of Chile, as shown in Figure 5A, has been used to describe the technique for removing cirrus absorption effects in the 11-μm BT image. Prior to cirrus removal, the mean temperature for the Figure 5E as well as Figure 5F) is 280.4 K. After obtaining the slope, *S*_BT_, we have made cirrus absorption corrections based on Equation (4). The resulting cirrus-removed 11-μm BT image is shown in Figure 5H, with the scene mean temperature of 284.0 K. The spatial variability of the cirrus-removed Figure 5H image is small. To be more quantitative, the standard deviation of Figure 5H scene temperatures is only 0.54 K. The Figure 5H image demonstrates clearly the effective removal of cirrus absorption effects in the original TOA 11-μm band BT image (see Figure 5F).

### 3.2. A Land/Water Boundary Scene, Maryland, USA, 17 April 2014

A portion of a Landsat 8 OLI RGB image acquired over a coastal area in the State of Maryland, USA, on 17 April 2014, is shown in Figure 6A. The scene contained solid land surfaces and muddy soil and sandy surfaces. The corresponding 1.375-μm band image is shown in Figure 6B, and the false-colored 11-μm band BT image in Figure 6C. The bottom half of the scene had slightly thick cirrus clouds (see Figure 6B) and therefore smaller BT values (see Figure 6C). Figure 6D is the 11-μm band BT image after the removal of cirrus absorption effects. By comparing Figure 6C with Figure 6D, it is seen that the solid land surface features, in particular the right half of the scene, are seen much better after cirrus corrections.

### 3.3. A Water Scene, Baltic Sea, 11 August 2015

Figure 7 shows the third case of correction of cirrus absorption effects in the 11-μm band TOA BT image. Figure 7A is the RGB image of the portion of the scene. The image was acquired on 11 August 2015 over Baltic Sea and during a peak chlorophyll blooming event. Figure 7B is the TOA 1.375 μm band reflectance image. The cirrus features were spatially diffused. Figure 7C is the TOA 11-μm band BT image. The thin cirrus-affected areas (in blue color) have smaller BT values. Figure 7D is the cirrus-effect-removed TOA 11-μm band BT image. The spatial distributions of BT values are far more uniform after the cirrus removal. The bottom curve in Figure 7E is the horizontal BT profile along the red line in Figure 7C (from left to right). The top curve in Figure 7E is the corresponding cirrus-corrected BT profile along the red line in Fig 7D (also from left to right). After the cirrus corrections, the BT values along the red horizontal line are well within ±1 K. By comparing the two curves in Figure 5E, it is seen that the maximum BT adjustment is about 15 K (near sample 100). The images and plots in Figure 7 demonstrate again that thin cirrus clouds can depress significantly the satellite-measured BT values, and corrections to measured BT values can be made with the empirical technique described in this article.

## 4. Discussion

The USGS operational Landsat 8/9 surface temperature (ST) data products are generated with a single-channel algorithm [5]. The algorithms require not only the 11-μm band TOA BT data as an input, but also many additional input data sets for constraining the retrievals. For example, the required input data sets include atmospheric temperature and pressure profiles, specific humidity, and air temperature extracted from reanalysis data [1]. The information about cirrus clouds, and more importantly the widespread thin cirrus, is not present in the reanalysis data. It is generally believed that the Landsat 8 ST product accuracy is about 1 K for very clear atmospheric conditions [10]. However, as shown in Figure 1 and Figure 2, thin cirrus clouds can easily depress the Landsat 8 Level 2 ST data products by 10 to 20 K. The thin cirrus cloud scattering effects are also directly propagated down to the Level 2 surface reflectance data products, particularly over dark water surfaces (see Figure 2C). The test cases here suggest that if the cirrus absorption effects were pre-removed from the TOA L1B 11-μm band BT images with the technique described here, and the cirrus-corrected 11-μm band BT images were then used in the Level 2 ST retrievals, the real accuracy of the Level 2 ST data products would be greatly improved. In a sense, the cirrus correction can serve as one important pre-processing step, particularly for spatially homogeneous thin cirrus scenes, prior to the normal ST retrievals with the USGS operational ST algorithm.

The focal plane assembly [1,25] of an OLI instrument is designed in such a way that, for a given surface pixel, different OLI bands image the pixel at different time and viewing angles. Although the USGS publicly released OLI L1B data products for a given surface pixel are spatially registered, the elevated water cloud and cirrus cloud features in different bands are not spatially registered. The cirrus feature shifts between the 11-μm band image and the 1.375-μm band image hinder the proper removal of cirrus absorption effects in 11-μm band images for spatially non-uniform cirrus cases with the technique described in this article. For MODIS, all bands view a surface pixel ‘simultaneously’, i.e., at the same time and the same viewing angle. Cirrus feature shifts among images of different MODIS bands are therefore not a problem. As a result, the technique described here is more suited for removing cirrus absorption features in MODIS IR band images. However, the coarse spatial resolution (approximately 1 km) of MODIS surface temperature data products and the associated ET data products are too coarse for applications requiring higher spatial resolution. The finer spatial resolution (approximately 100 m or less) Landsat ET products can be more usable for many practical applications, such as irrigation scheduling for farm fields [13] and the study of urban heating effects [27].

The empirical method for finding a straight fitting line in a scatter plot between images of two L8 bands, such as the one illustrated in Figure 5C, and for the subsequent correction of cirrus scattering effects in L8 bands below 2.5 μm, was developed on a Mac desktop computer in around 2015. The algorithm was written in Fortran 90, and contained no direct input and output routines to read the publicly released ‘standard’ Landsat 8 data sets. We used other available software packages to extract relevant images and geolocation information from the standard Landsat 8 data, and stored the extracted data in plain binary files. We then used the Fortran 90 routines to read in the input binary files, and also output the cirrus-removed data into plain binary files. The previous F90 routines were modified recently on a MacBook Pro computer to perform cirrus corrections in BT images of the 11-μm band. In summary, at present, our Landsat cirrus correction algorithms are not implemented onto any computer systems to perform ‘operational’ processing of Landsat 8/9 data. However, the key portion of the F90 codes (or slight variations of the codes) for finding the straight fitting line (see Figure 5C) is also contained in the NASA operational MODIS and VIIRS cirrus reflectance algorithms [23,24] that are publicly available from a NASA data center. In principle, both algorithms can be modified to perform ‘operational’ cirrus corrections for Landsat 8/9 data. 

## 5. Summary

Because cirrus clouds are generally formed in the upper part of the troposphere (approximately 10 km) and the lower part of the tropopause, where the air temperatures are very low (approximately 220 K), the presence of a layer of thin cirrus clouds can cause a big depression in satellite-measured 11-μm band BT values. In Section 2, we provided a literature review on BT sensitivities to thin cirrus clouds. Also in Section 2, we detailed the steps on using the correlations between the 1.375-μm cirrus band image and the 11-μm band TOA BT image for practical removal of cirrus absorption effects in the 11-μm band BT image. In Section 3, we presented sample corrections to cirrus-affected BT images for spatially homogeneous cases with an accuracy of ±1 K. We suggest that the cirrus-removed TOA 11-μm band BT images should be used for the retrieval of the Level 2 surface temperature (ST) data products. 

## Figures and Tables

**Figure 1 sensors-24-04697-f001:**
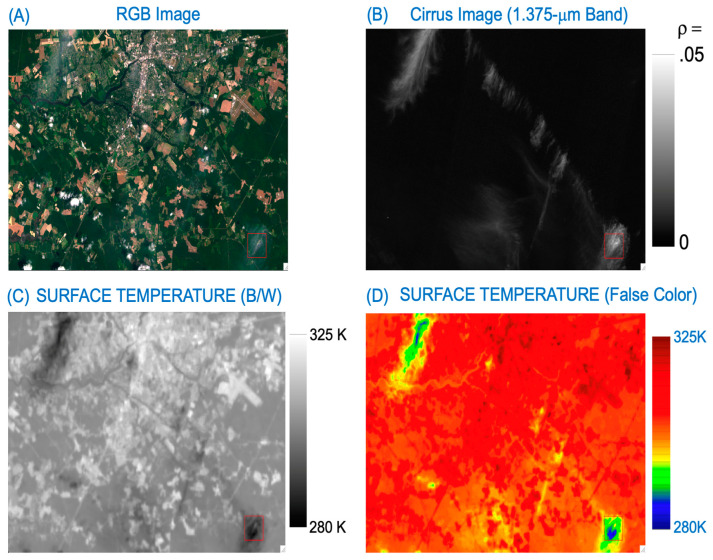
(**A**) A Landsat 8 OLI RGB image (Path: 014; Row: 033) acquired over eastern coastal area of Maryland State, USA, on 1 July 2018, (**B**) the corresponding Band 9 (cirrus band) image, (**C**) the surface temperature image, and (**D**) the false-color-coded surface temperature image.

**Figure 2 sensors-24-04697-f002:**
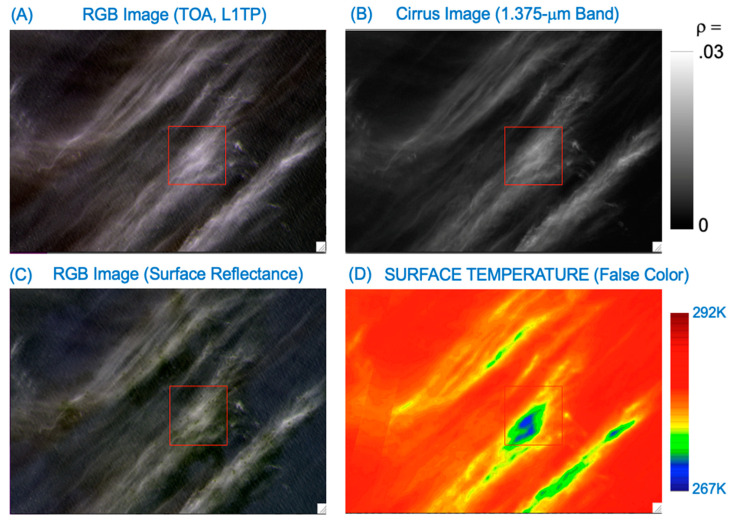
(**A**) A Landsat 8 OLI RGB image (Path: 001; Row: 085) acquired on 5 March 2014, over Pacific Ocean west of the coastal area of Chile, (**B**) the corresponding Band 9 (cirrus band) image, (**C**) the Level 2 OLI RGB surface reflectance (SR) image, and (**D**) the false-color-coded water surface temperature image with an attached color bar.

**Figure 3 sensors-24-04697-f003:**
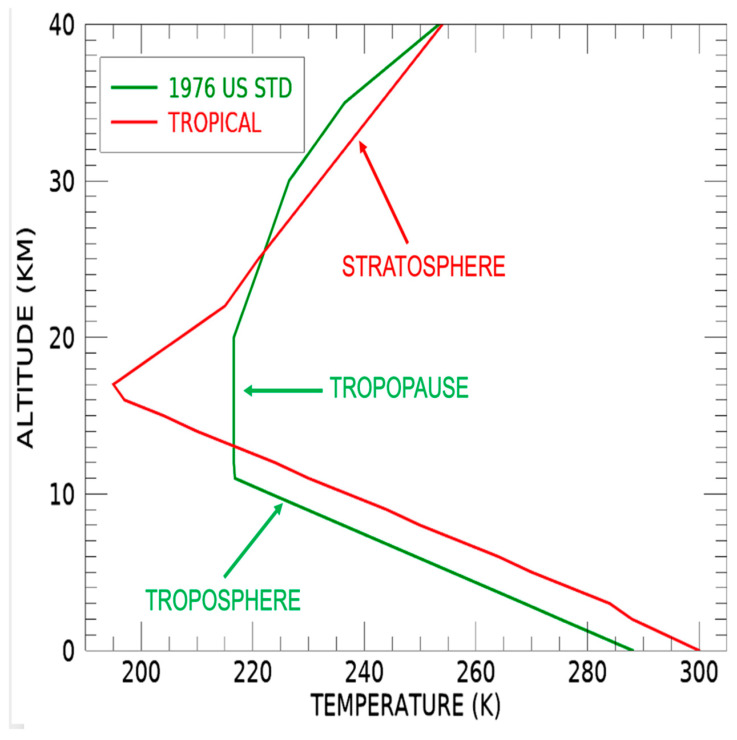
Vertical temperature profiles for the Tropical and U.S. Standard (1976) model atmospheres.

**Figure 4 sensors-24-04697-f004:**
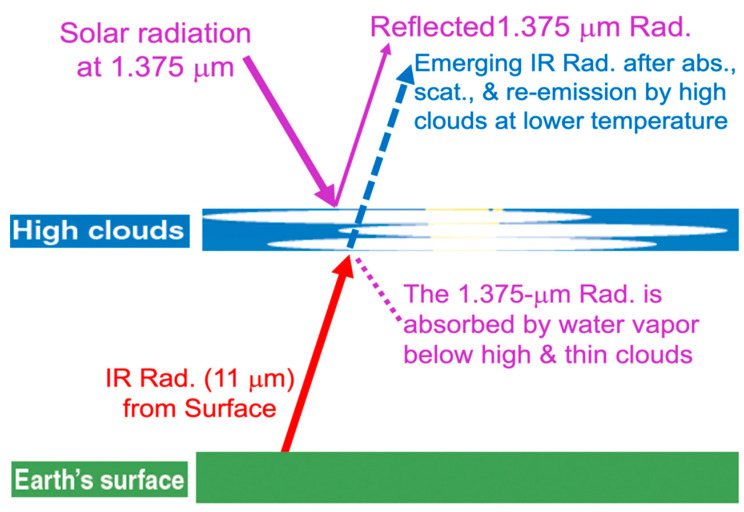
A diagram illustrating that the IR radiances originating from Earth’s surface are absorbed, scattered, and re-emitted at lower temperatures by cirrus clouds at high altitudes (approximately 10 km) and that the downward solar radiation at 1.375-μm is scattered backup by cirrus clouds and the transmitted 1.375-μm radiance through cirrus is absorbed by atmospheric water vapor beneath cirrus clouds.

**Figure 5 sensors-24-04697-f005:**
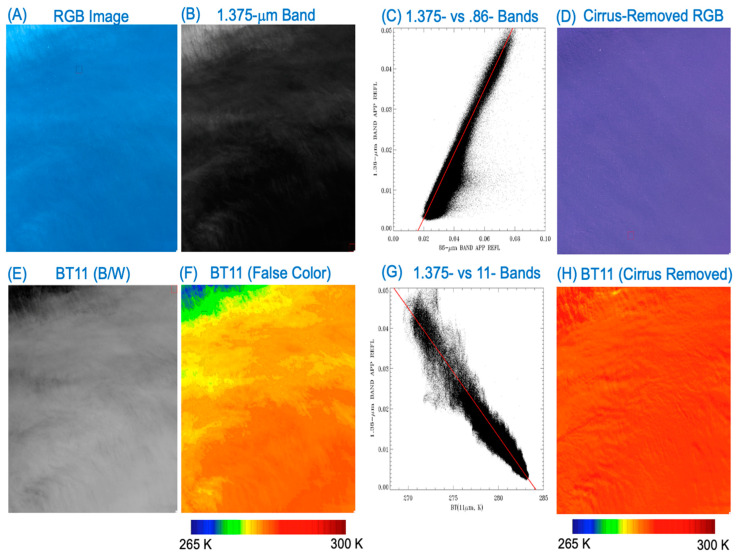
(**A**) A Landsat 8 OLI RGB image (Path: 001; Row: 085) acquired on 21 March 2014, over Pacific Ocean west of the coastal area of Chile, (**B**) the corresponding Band 9 (cirrus) image, (**C**) the scatter plot of 1.375-μm versus 0.86-μm band images, (**D**) the cirrus-corrected RGB image, (**E**) the black/white 11-μm BT image, (**F**) the false-colored version of 11-μm BT image, (**G**) the scatter plot of 1.375-μm band TOA reflectance image versus the 11-μm band TOA BT image, and (**H**) the 11-μm band BT image after the removal of cirrus absorption effects.

**Figure 6 sensors-24-04697-f006:**
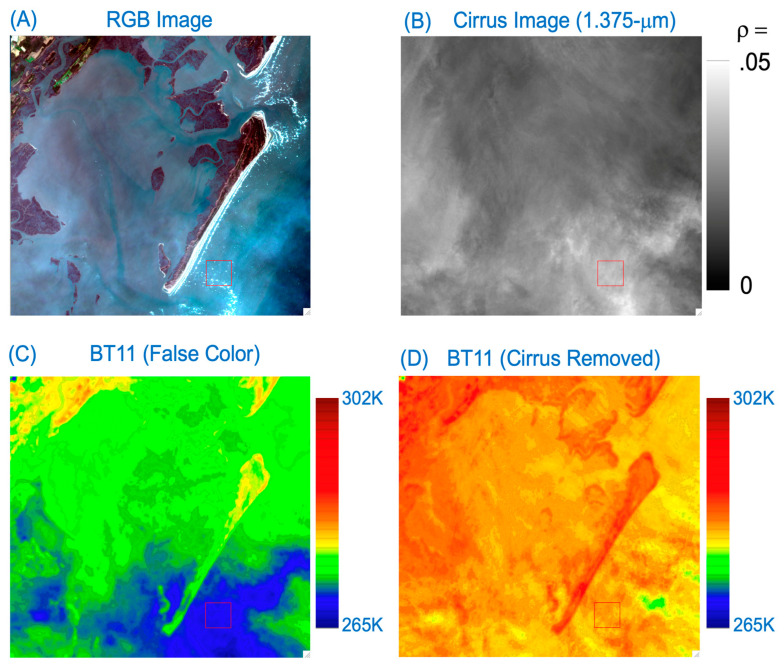
(**A**) A portion of Landsat 8 OLI RGB image (Path: 014; Row: 034) acquired on 17 April 2014, over eastern coastal area of Maryland, USA, (**B**) the corresponding Band 9 (cirrus band) image, (**C**) the false-colored version of 11-μm band BT image, and (**D**) the 11-μm band BT image after the removal of cirrus absorption effects.

**Figure 7 sensors-24-04697-f007:**
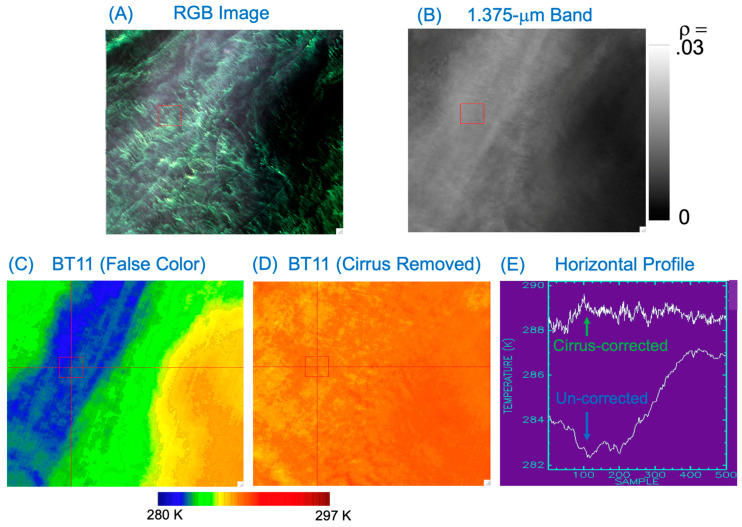
(**A**) A portion of a Landsat 8 OLI RGB image (Path: 190; Row: 019) acquired on 11 August 2015 over the Baltic Sea, (**B**) the Band 9 (cirrus band) image, (**C**) the false-colored version of 11-μm band BT image, (**D**) the 11-μm band BT image after the removal of cirrus absorption effects, (**E**) the horizontal BT profiles along the red-colored line in (**C**) (bottom curve), and the cirrus-corrected BT profile along the same line in (**D**) (top curve).

**Table 1 sensors-24-04697-t001:** Landsat 8/9 OLI and TIRS band names, widths, and spatial resolution (m).

Bands	Wavelength (μm)	Resolution (m)
Band 1–Ultra Blue	0.43–0.45	30
Band 2–Blue	0.45–0.51	30
Band 3–Green	0.53–0.59	30
Band 4–Red	0.64–0.67	30
Band 5–Near-Infrared (NIR)	0.85–0.88	30
Band 6–Shortwave Infrared (SWIR) 1	1.57–1.65	30
Band 7–Shortwave Infrared (SWIR) 2	2.11–2.29	30
Band 8–Panchromatic	0.50–0.68	15
Band 9–Cirrus	1.36–1.39	30
Band 10–Thermal Infrared (TIRS) 1	10.6–11.19	100
Band 11–Thermal Infrared (TIRS) 2	11.5–12.51	100

## Data Availability

The Landsat8 data used in this study is downloaded from an USGS web site through a link provided in [2]. The processed data is available upon request.

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
