# Peer review of "Correction of Thin Cirrus Absorption Effects in Landsat 8 Thermal Infrared Sensor Images Using the Operational Land Imager Cirrus Band on the Same Satellite Platform"

_sensors, 2024, doi:10.3390/s24144697_

Round 1

Reviewer 1 Report

Comments and Suggestions for Authors

Dear Authors,

        The manuscript “Correction of thin cirrus absorption effects in Landsat 8 TIRS Images using the OLI Cirrus Band on the same satellite platform” the highlight of the paper is the presentation of three homogeneous thin cirrus case studies focusing on removing cirrus absorption effects in the 11-μm band BT images. The research addresses the challenges in removing cirrus absorption effects due to mismatching of cirrus spatial features in different bands.

The manuscript highlights the importance of Landsat data, together with MODIS data, for various applications such as estimating evapotranspiration (ET) for water management, vegetation monitoring, drought detection, yield prediction, and irrigation scheduling. The use of Landsat ST data products for ET estimates may lead to significant errors due to thin cirrus contamination, emphasizing the necessity of effective cirrus removal techniques.

With regard to the manuscript, specific corrections will be necessary, which have been duly noted in the comments on the digital file. Please check the punctuation, especially the use of commas.

I’d like to draw your attention to the following comments:

1) Revise the abstract: what is the problem addressed in the study? What alternative approach was used? What improvement was achieved? Cite the sites where the tests were carried out.

2) Cite the reference for Figure 3 and Figure 4.

3) Cite the Path and Row of the Landsat scenes used in the study.

4) Inform the repository of the Landsat images: EarthExplorer or Glovis?

5) Mention the hardware configuration and applications used to develop the codes.

6) Will the code developed be made available on github? If so, please inform.

7) A flowchart showing the processing steps would be beneficial for understanding the experiment carried out.

8) It would be interesting to compare the results obtained by processing the same images in widespread applications: 1) atmospheric correction tools such as the Landsat Surface Reflectance Code (LaSRC) to adjust for atmospheric effects. These tools often include options to correct for cirrus clouds. 2) Use cloud masking algorithms like Fmask (Function of mask) which is included in tools like Google Earth Engine. These algorithms can automatically detect and mask clouds, including cirrus.

9) Is the analysis of only three parts of scenes enough to affirm that the algorithm developed is efficient? This is a point that should be carefully substantiated.

10) Adequately answer the objective of the study presented in chapter 1.

I conclude by congratulating you on your study.

Respectfully,

Comments on the Quality of English Language

Punctuation should be revised, especially the use of commas. Avoid using long paragraphs as they tend to make reading confusing. I have indicated in the comments of the digital file the places where the writing should be improved. When introducing acronyms, their meaning should be presented.

Author Response

The manuscript “Correction of thin cirrus absorption effects in Landsat 8 TIRS Images using the OLI Cirrus Band on the same satellite platform” the highlight of the paper is the presentation of three homogeneous thin cirrus case studies focusing on removing cirrus absorption effects in the 11-μm band BT images. The research addresses the challenges in removing cirrus absorption effects due to mismatching of cirrus spatial features in different bands.

The manuscript highlights the importance of Landsat data, together with MODIS data, for various applications such as estimating evapotranspiration (ET) for water management, vegetation monitoring, drought detection, yield prediction, and irrigation scheduling. The use of Landsat ST data products for ET estimates may lead to significant errors due to thin cirrus contamination, emphasizing the necessity of effective cirrus removal techniques.”

– Thanks for spending your valuable time to review our manuscript and to provide valuable comments!

With regard to the manuscript, specific corrections will be necessary, which have been duly noted in the comments on the digital file. Please check the punctuation, especially the use of commas.

I’d like to draw your attention to the following comments:

1) Revise the abstract: what is the problem addressed in the study? What alternative approach was used? What improvement was achieved? Cite the sites where the tests were carried out.” – In responding to this comment and the comment from another reviewer, we removed a few sentences from the abstract to make it shorter. Specifically, we removed the sentences “The TIRS instrument has two IR bands centered near 11 and 12 micron, respectively. Temperature retrievals are made from the Level 1 top of atmosphere (TOA) products, such as the 11-micron band brightness temperature (BT), the TOA reflectances, in addition to ancillary data that include Normalized Difference Vegetation Index (NDVI), surface emissivity, atmospheric profiles obtained from other sources” from abstract. As for ‘what is the problem’ – no change is made, because we already stated in the abstract that “thin cirrus cloud features present in the Level 1 1.375-micron band images are directly propagated down to the Level 2 surface data products. The surface temperature errors resulting from thin cirrus contamination can be 10 K or larger”. As for ‘What alternative approach was used’ – no modification in the abstract is made. Basically, in the present USGS operational surface temperature retrieving algorithm, thin cirrus effects have not been addressed at all, but thick cirrus pixels were masked in the L8/9 cloud masking data product. The algorithm uses model simulated atmospheric temperature profile, and tries to model out atmospheric effects. However, thin cirrus clouds are not present in the model-simulated atmospheric temperature profiles. As for ‘What improvement was achieved’ – we already have the words “can be reduced to about 1 K” (from 10 K or larger).

2) Cite the reference for Figure 3 and Figure 4”. – We produced the two figures ourself. The data source for Fig. 3 was cited in Reference [16]. We draw Fig. 4 originally for use in our  submitted NASA ECOSTRESS proposal in 2022, and to help proposal reviewers to have a rough idea about ‘cold’ (in blue color) ice particle absorption effect.

3) Cite the Path and Row of the Landsat scenes used in the study.” – We added the path and row numbers to all figure descriptions.

4) Inform the repository of the Landsat images: EarthExplorer or Glovis?” – we decided not to do so, such as providing a web link to EarthExplorer. The web links provided in References 1 and 2 should serve as good starting places for people to get access to USGS Landsat 8/9 data. Over the years, the USGS web sites changed quite a bit. For examples, in the 2015 – 2016 time frame, we figured out a way to get access to Landsat 8 data. By 2023, the old way to get the data was no longer working, and we spent time again to figure out a new way to download Landsat 8/9 data. By May of 2024, we needed to make adjustment again in order to get re-processed Landsat 8/9 L1 and L2 data products. By the way, we also found that, through a straight forward google search, we could get updated web links for Landsat data access from USGS.

“5) Mention the hardware configuration and applications used to develop the codes.

6) Will the code developed be made available on github? If so, please inform.

7) A flowchart showing the processing steps would be beneficial for understanding the experiment carried out.”

- We are responding to the above 3 comments. Basically, we added a new paragraph in the “Discussion Section”. The added content is

“The empirical method for finding a straight fitting line in a scatter plot between images of two L8 bands, such as the one illustrated in Fig. 5C, and for subsequent correction of cirrus scattering effects in L8 bands below 2.5 micron was developed on a Mac desktop computer around 2015. The algorithm was written in Fortran 90, and contained no direct input and output routines to read the publicly released ‘standard’ Landsat 8 data sets. We used other available software packages to extract relevant images and geolocation information from the standard Landsat 8 data, and stored the extracted data into plain binary files. We then used the Fortran 90 routines to read in the input binary files, and also output the cirrus-removed data into plain binary files. The previous F90 routines were modified recently on a macbook pro computer to do cirrus corrections in BT images of the 11-micron band. In summary, at present, our Landsat cirrus correction algorithms are not implemented onto any computer systems to do ‘operational’ processing of Landsat 8/9 data.

However, the key portion of the F90 codes (or slight variations of the codes) for finding the straight fitting line (see Fig. 5C) is also contained in the NASA operational MODIS and VIIRS cirrus reflectance algorithms [23, 24] that are publicly available from a NASA data center. In principle, both algorithms can be modified to do ‘operational’ cirrus corrections for Landsat 8/9 data”. It takes research funding and manpower to do so (just as we did for help in implementing an operational hyperspectral atmospheric correction algorithm on a NASA JPL computer in 2012, and JPL has been using the software to do AVIRIS atmospheric corrections since that time, and publicly released the atmosphere-corrected AVIRIS data).

More historically speaking – around 2015, we spoke with an USGS Landsat 8 manager (during a NASA HysPRI conference) regarding cirrus corrections for L8 bands below 2.5 micron. He was very interested on the cirrus correction issue. Soon after the conversation, we modified a version of our MODIS cirrus reflectance algorithm codes to do Landsat 8 data processing. We used ENVI software package to extract plain binary images from the ‘standard’ L8 data sets, and we found an error in ENVI’ packages internal coding in dealing with the add and offset terms. Then, with the help from other people, we wrote an IDL version of code to do data extraction with proper applications of the gain and offset terms. With the extracted L8 data, we input the data to our L8 version of cirrus reflectance algorithm, and processed data for about 10 selected scenes. We e-mailed our results to the Landsat 8 manager in South Dakota, and he never responded to our e-mail. At the time, we hoped to get funding from USGS for integrating our L8 cirrus reflectance algorithm to the L8 operational computing environment. Subsequently, we spoke with two other USGS Landsat managers (one at USGS in Reston, Virginia and one in South Dakota). No positive outcomes came out. In general, to produce an operational computing software package, it takes at least 3 times more man hours than to produce a just functional version of the software package, based on our experiences with developing the operational versions of MODIS and VIIRS cirrus reflectance algorithms.

The idea of putting not well tested computer onto ‘github’ is actually pretty bad. Any software packages for public uses need to have financial and manpower support behind the scene. At our research institution, we are more in the business of fee for service, and we do not provide free public support. The code developed through basic research funding, if needs to be transitioned to operational use, will be delivered to another operational entity for code integration and testing. After passing through all testing, the updated algorithm can then be put into a computing environment for ‘operational’ data processing. In this way, the code will always be supported (such as after computer upgrades).

The idea of putting not well tested (research grade) computer code onto ‘github’ for public use is naïve. In my specific case, I delivered a software package to NASA GSFC for fast calculation of spectral atmospheric corrections about 10 years ago. GSFC decided to add a bit of descriptions to the code, and placed the code for public access. GSFC didn’t provide funding for my continued support of the software package. Unfortunately, GSFC people themselves were unable to use the code correctly for simulating the pre-launch PACE OCI data in the 2020 and 2021 time frame. They didn’t figure out a way to deal with pixel-by-pixel variations in solar and view angles for the very large swath OCI data (~3000 km).

8) It would be interesting to compare the results obtained by processing the same images in widespread applications: 1) atmospheric correction tools such as the Landsat Surface Reflectance Code (LaSRC) to adjust for atmospheric effects. These tools often include options to correct for cirrus clouds. 2) Use cloud masking algorithms like Fmask (Function of mask) which is included in tools like Google Earth Engine. These algorithms can automatically detect and mask clouds, including cirrus”. – As stated above, at NRL in DC, we are not supposed to release computer code developed through basic research funding project publicly. Also, we are not allowed accessing to instant data sharing services, such as ‘google drive’ and ‘google engine’. If we hope to release codes as well as manuscripts to the public domain, everything needs to go through a ‘publication release process’. I will not say anything publicly about LaSRC and Fmask. However, since USGS released L8/L8 L1 and L2 data publicly, it is very easy for people to see the cirrus contamination problems in the L2 surface reflectance (SR) and temperature (ST) data products. Unlike implementing NDVI algorithm into Google Earth Engine, the porting over of our F90 code into the Google Earth Engine environment would not be a simple task. The thin cirrus clouds we talked about in the present manuscript generally are not masked out in cloud masking algorithms, and we noticed this fact in both the Landsat 8/9 surface temperature data product and in the NASA ECOSTRESS surface temperature data product.

“9) Is the analysis of only three parts of scenes enough to affirm that the algorithm developed is efficient? This is a point that should be carefully substantiated.” – We decided just to show 3 cases as examples in the present manuscript. So far, we processed about 20 Landsat 8/9 scenes over different parts of the earth. A few days ago, we noticed that, the surface temperature values over Arctic water pixels contaminated by thin cirrus clouds can be totally wrong.  For the water pixels, the USGS reported surface temperature values were well below the water freezing point (~273.15 K).

“10) Adequately answer the objective of the study presented in chapter 1.” We already answered earlier.

Reviewer 2 Report

Comments and Suggestions for Authors

1. A brief summary

The article "Correction of thin cirrus absorption effects in Landsat 8 TIRS Images using the OLI Cirrus Band on the same satellite platform" touches on a very relevant topic. The authors demonstrate a deep understanding of the problem and propose to apply a previously proven approach to cirrus correction onto thermal channels of Landsat-type sensors. The manuscript is written very concisely and competently, the results and activities of the authors deserve attention and respect. Such a work should certainly be published. Below are some minor comments that may help improve the article made by the honored masters.

2. General concept comments

2.1 It is unclear why Landsat 9 is not mentioned in the title of the article. As far as I understand, the proposed data preprocessing technique is also applicable to data from this satellite. At the same time, it is unclear why the authors did not experiment with the Landsat 9 data. I think a comment on this point should be placed in the article.

2.2 The authors conducted only 3 experiments. I would recommend consider and analyze more examples before using the algorithm for mass application and updating Landsat data collections. A larger number of examples would probably allow us to discover additional features of the output product, which will subsequently make the product more usable.

3. Specific comments

3.1 In order to comply with academic rigor, it is desirable to support more statements with references, for example, a reference is required in line 181.

3.2. In the text and in the figures, there are different versions of the description of band 9 (1.375 micrometers, 1.38 micrometers, or as a band with a range of 1.36-1.39 micrometers). Given the highest level of publication, it is advisable to observe academic rigor and achieve uniformity. In particular, it is desirable to add central wavelengths column to the wavelength table.

4. Typos

The symbol "micrometers" is somewhere incorrectly displayed (for example, see the annotation).

Thank you so much for the interesting work!

Author Response

  1. A brief summary

“The article "Correction of thin cirrus absorption effects in Landsat 8 TIRS Images using the OLI Cirrus Band on the same satellite platform" touches on a very relevant topic. The authors demonstrate a deep understanding of the problem and propose to apply a previously proven approach to cirrus correction onto thermal channels of Landsat-type sensors. The manuscript is written very concisely and competently, the results and activities of the authors deserve attention and respect. Such a work should certainly be published. Below are some minor comments that may help improve the article made by the honored masters.”

- Thanks for spending your valuable time to review our manuscript and for the very positive comments!

  1. General concept comments

2.1 It is unclear why Landsat 9 is not mentioned in the title of the article. As far as I understand, the proposed data preprocessing technique is also applicable to data from this satellite. At the same time, it is unclear why the authors did not experiment with the Landsat 9 data. I think a comment on this point should be placed in the article.” – We understand your fine point, and that the Landsat 8/9 have basically identical instruments. Due to the short time period before submitting the revised manuscript, we do not have enough time to produce nice-looking images and results from Landsat 9 data sets and to put them in the revised manuscript. On the other hand, the addition of new cases does not enhance our ability for illustrating the cirrus-removing technique from the 11-micron band BT images.

2.2 The authors conducted only 3 experiments. I would recommend consider and analyze more examples before using the algorithm for mass application and updating Landsat data collections. A larger number of examples would probably allow us to discover additional features of the output product, which will subsequently make the product more usable.” -  We decided just to show 3 cases as examples in the present manuscript. So far, we processed about 20 Landsat 8/9 scenes over different parts of the earth. A few days ago, we noticed that, the surface temperature values over Arctic water pixels contaminated by thin cirrus clouds can be totally wrong:  for the water pixels, the USGS reported surface temperature values were well below the water freezing point (~273.15 K). On the other hand, we applied our BT11 cirrus correction algorithm, as described in the present manuscript, to four L9 scenes over Alaska (including the LC09_L1TP_010058_20240510 data set), we found out that the BT11 cirrus correction method worked fine. However, in the corresponding LC09_L2SP_010058_20240510 data set, the USGS reported surface temperatures over water pixels were lower than 273 K, in some cases lower than 250 K. I would like to speculate that the assumed atmospheric vertical temperature profile used in the surface temperature retrievals over Arctic regions (when thin cirrus were present) was not right.

  1. Specific comments

3.1 In order to comply with academic rigor, it is desirable to support more statements with references, for example, a reference is required in line 181.” – no new reference is added in. The fine point is contained in the cited reference [16], but not easy to catch. On the other hand, although US 1976 model atmosphere was used in numerous publications, I was told in mid-1980s that the US 1976 model atmosphere’s vertical pressure, temperature, and volume mixing ratios of water vapor violated the atmospheric thermal dynamic principles, while the US1962 model atmospheres were properly constructed (no violation of thermal dynamics and radiation).

3.2. In the text and in the figures, there are different versions of the description of band 9 (1.375 micrometers, 1.38 micrometers, or as a band with a range of 1.36-1.39 micrometers). Given the highest level of publication, it is advisable to observe academic rigor and achieve uniformity. In particular, it is desirable to add central wavelengths column to the wavelength table.” – The words ‘1.38’ in the text, Figure 4 and descriptions, were changed to ‘1.375’. However, for historical reasons, in the cited references, there were ‘1.38’ in paper titles. The difference of 5 nm actually came from our original specification for MODIS band 26 (1.36 – 1.39 micron) and the actual manufactured filter peak response curve, which deviated by about 5 nm from the original specification. We decided not to add the Landsat 8 OLI central wavelengths to the wavelength table. If we add the originally specified wavelength center positions to the table, the actual peak transmission positions can deviate from the original specifications. The filter peak positions can also change a bit with time during the vacuum space environment. On the other hand, even if the filter responses are shifted, the transmission range would still remain about the same (I mean not becomes out of the ranges in comparison with those in the original specifications).

  1. Typos

‘The symbol "micrometers" is somewhere incorrectly displayed (for example, see the annotation).’ –  We searched through the MS version of the original manuscript, and didn’t find the mis-spelling.

Reviewer 3 Report

Comments and Suggestions for Authors

In the abstract, (1) it is a bit too long in its current form. Can the authors shorten it? Additionally, please present the findings in numerical order, e.g., (1), (2), etc.

The paper is very interesting as it demonstrates a good use of Landsat imagery for surface temperature retrievals. However, it lacks suggestions for using these remote sensing products in other aspects. For example, after L36 [consider shortening the abstract and including an additional suggestion to use this data after improved using the proposed method]. Besides, between L374-376, as we know, temperature retrieval using satellite imagery is mostly used for observing changes in surface temperature, especially for urban heat. I suggest the authors include a short discussion on how the findings of this study could be better utilized to replace traditional ground-truth data for urban heat observation in cities (suggest doi.org/10.3390/w14213367).

Despite the experiment being well conducted and the results clearly presented, can the authors provide limitations of this proposed method?

Is it necessary to test with a higher number of areas to demonstrate this method’s reliability?

The main findings mentioned in the conclusion need to be labeled numerically, e.g., (1), (2), etc.

Author Response

Thanks for your time and efforts in reviewing our manuscript!

In the abstract, (1) it is a bit too long in its current form. Can the authors shorten it? Additionally, please present the findings in numerical order, e.g., (1), (2), etc.” - we removed the following sentences from the abstract. They are: “The TIRS instrument has two IR bands centered near 11 and 12 mm, respectively. Temperature retrievals are made from the Level 1 top of atmosphere (TOA) products, such as the 11-mm band brightness temperature (BT), the TOA reflectances, in addition to ancillary data that include Normalized Difference Vegetation Index (NDVI), surface emissivity, atmospheric profiles obtained from other sources.”

The paper is very interesting as it demonstrates a good use of Landsat imagery for surface temperature retrievals. However, it lacks suggestions for using these remote sensing products in other aspects. For example, after L36 [consider shortening the abstract and including an additional suggestion to use this data after improved using the proposed method]. Besides, between L374-376, as we know, temperature retrieval using satellite imagery is mostly used for observing changes in surface temperature, especially for urban heat. I suggest the authors include a short discussion on how the findings of this study could be better utilized to replace traditional ground-truth data for urban heat observation in cities (suggest doi.org/10.3390/w14213367).” – The suggested reference is added into the citation.

Despite the experiment being well conducted and the results clearly presented, can the authors provide limitations of this proposed method?” – No change is made to the manuscript. So far, we have not found cases that the method does not work for correcting thin cirrus clouds in the 11-micron BT images. The method is based on solid physics and atmospheric spectroscopy principles.

Is it necessary to test with a higher number of areas to demonstrate this method’s reliability?” – Actually, we tested about a total of about 20 cases of Landsat 8 and 9 scenes, but we now do not have enough time to generate high quality figures for inclusion in this version of revised manuscript. Please also see our lengthy answers to comments (2.2 The authors conducted …) from the 2nd reviewer.

The main findings mentioned in the conclusion need to be labeled numerically, e.g., (1), (2), etc.” – Since we do not have many findings, there is no need to add numerical labels.

Round 2

Reviewer 1 Report

Comments and Suggestions for Authors

Dear Authors,

      Thank you for sending us the new version of the manuscript, together with the cover letter detailing the revisions made. I appreciate the care and attention you have given to the suggestions.

      I have carried out a detailed analysis of the new version of the manuscript, comparing it thoroughly with the previous version. I was pleased to see that the suggestions were adequately implemented or justified and contributed significantly to improving the quality of the work.

      Once again, thank you for your commitment and cooperation. 

Respectfully

Reviewer 3 Report

Comments and Suggestions for Authors

Thank you for the revision.